# Clinical Efficacy of a Single Intravenous Regional Limb Perfusion with Marbofloxacin versus Ceftiofur Sodium to Treat Acute Interdigital Phlegmon in Dairy Cows

**DOI:** 10.3390/ani13101598

**Published:** 2023-05-10

**Authors:** Gianluca Celani, Paola Straticò, Paolo Albano, Lucio Petrizzi, Carlo Maria Mortellaro, Vincenzo Varasano

**Affiliations:** 1Veterinary Teaching Hospital, Department of Veterinary Medicine, University of Teramo, 64100 Teramo, Italy; pstratico@unite.it (P.S.); lpetrizzi@unite.it (L.P.); vvarasano@unite.it (V.V.); 2Bovine Practitioner, 85100 Potenza, Italy; 3Department of Veterinary Sciences and Public Health, University of Milan, 20133 Milan, Italy; carlomaria.mortellaro@unimi.it

**Keywords:** cattle, interdigital phlegmon, intravenous regional limb perfusion, antimicrobial

## Abstract

**Simple Summary:**

Bovine interdigital phlegmon (IP) is an infective bacterial disease that originates from a lesion in the interdigital skin that rapidly spreads into the deeper soft tissues of the foot. Intravenous regional limb perfusion (IVRLP) with antimicrobials is a well-known technique that provides high antimicrobial concentrations at the site of infection, including soft tissue, bones, joints and tendon sheaths, without the need for systemic administration. The results of this field study support the clinical impression that antimicrobial IVRLP is an efficacious procedure for treating IP in dairy cows.

**Abstract:**

The objective of the study was to compare the clinical efficacy of a single antimicrobial intravenous regional limb perfusion (IVRLP) with marbofloxacin versus ceftiofur sodium to treat naturally occurring interdigital phlegmon (IP) in dairy cows. The study had a randomized parallel-group design. Forty lactating Friesian cows clinically diagnosed with acute IP were enrolled, assigned to one of two treatment groups, and received a single IVRLP with the antimicrobial drug selected (M: 0.67 mg/kg of marbofloxacin; C: 500 mg/animal of ceftiofur sodium). Clinical data for the severity of lameness, digital swelling and local lesion appearance were assessed at diagnosis and at 5, 10 and 15 days post-IVRLP. Clinical resolution was defined as digital swelling disappearance, locomotion score reduction of at least 2/5 points, healed or healing local lesion and no relapse at 15 days after IVRLP. The total daily milk yield of each cow on the day before the clinical detection, on the day of diagnosis and on the day of the clinical follow-up post-IVRLP were registered. Lameness, digital swelling and local lesion severity were not significantly different between groups at any time-point. In both groups, on the fifteenth day after treatment, 17/20 (85%) cows showed a positive outcome, with no significant difference (*p* > 0.05). The daily milk production of all cows was adversely affected by the clinical onset of IP and gradually returned to a normal level after IVRLP in both groups. These preliminary results support the hypothesis that a single antimicrobial IVRLP procedure, irrespective of the antimicrobial selected (ceftiofur vs. marbofloxacin), has a high success rate and restores milk yield in cases of dairy cattle with acute IP lameness.

## 1. Introduction

Interdigital phlegmon (IP) has been known for centuries and is reported from several countries around the world [1,2]. This disease has also been referred to as foot rot, foul in the foot, interdigital necrobacillosis, lure, panaritium and interdigital or infectious pododermatitis [1,2]. IP is painful, causes lameness, is a costly disease and has a serious impact on farm profitability [3].

Typically, the condition occurs as a sporadic disease and can affect dairy and feedlot cattle. The overall incidence is probably less than 5%, but in epidemic outbreaks the incidence of the disease can be as high as 32% of the milking cows in a herd [4,5,6]. The Italian prevalence is 0.73% (46 herds and 3559 cows) [7], with an incidence of 20.6% being reported in a herd of 321 dairy cows [8].

Bovine IP is an infective bacterial disorder that originates from a lesion in the interdigital skin and rapidly extents into the deeper dermal layer of the foot. Multiple anaerobic organisms are likely to be synergistically involved in the pathogenesis. *Fusobacterium necrophorum*, *Porphyromonas levii* and *Prevotella intermedia* are common bacterial isolates from affected tissues [9,10].

The condition is characterized by inflammation and secondary soft tissue necrosis of the interdigital space. Usually, a sudden lameness occurs in only one limb in the affected cow, and the pelvic feet may be involved more often than the thoracic feet. The earliest clinical signs are erythema and symmetric swelling of the interdigital space and the coronary band. Progression of swelling results in separation of the claws, and the inflammatory oedema may extend to the pastern and fetlock regions. Soft tissue swelling leads to the development of foul-smelling necrotic cutaneous fissures within a few days. Blind foot rot or blind foul are terms used to indicate cases of IP in absence of an evident interdigital skin lesion, and this condition probably represents IP in an early stage of the disease [2,5].

If IP is diagnosed and treated early, the antibiotic treatment is usually successful, and most cases respond rapidly, whereas the response to delayed treatment can fail to control the infection, and often the cow has to be culled [2,5,10]. In more advanced IP cases, careful surgical debridement of necrotic tissue in the interdigital skin and bandaging of the infected foot are beneficial [5]. Possible secondary complications associated with IP may result in deep sepsis of the digit; this condition includes septic arthritis of the distal interphalangeal joint, septic tenosynovitis of the digital flexor tendon sheath, septic bursitis of the *bursa podotrochlearis*, abscess in the retroarticular space and septic osteitis or osteomyelitis of the distal phalangeal and navicular bone. The so-called super foul is a peracute form of IP in which development of deep sepsis can occur within two days of the onset of the clinical sign [5]. In such cases a more profound surgical procedure (digital amputation or tendon resection) may be used to save valuable cows [5].

Treatment of IP remains one of the main motives for therapeutic use of antimicrobials in cattle in Europe and USA. Administration of non-steroidal anti-inflammatory drugs (NSAIDs) is an optional adjunct therapy that helps to treat the inflammation and the pain associated with IP [2,5]. Several antimicrobial drugs have been used for the systemic treatment of IP, including procaine G penicillin (22,000–44,000 IU/kg IM SID or BID), amoxicillin (6.6 to 11 mg/kg IM or SC for 5 days), oxytetracycline (10 mg/kg IM or SC), sulfadimethoxine (55 mg/kg PO or IV loading, then 27.5 mg/kg PO or IV for 5 days), erythromycin (2.2 to 4.4 mg/kg IM daily for up to 5 days), ceftiofur hydrochloride/sodium (1.1 to 2.2 mg/kg SC or IM SID for 3 to 5 days), tylosin (18 mg/kg IM SID for up 5 days), sulfamethazine (30 g/100 kg PO, repeated in 72 h), florfenicol (40 mg/kg SC once), tulathromycin (2.5 mg/kg SC once), ceftiofur crystalline free acid (6.6 mg/kg SC in the base of the ear once) and tilmicosin (5 mg/kg SC) [2,11,12]. However, in the case of individual therapy for IP, clinical evidence to support the efficacy of antimicrobial parenteral treatment is only available for florfenicol, ceftiofur, tulathromycin and oxytetracycline [11]. A spontaneous resolution of IP is possible; however, the risk of secondary complications and the recovery time are increased.

Antimicrobial susceptibility testing results for cattle’s IP isolates of *Fusobacterium necrophorum* are lacking in the scientific literature. Furthermore, a higher likelihood of success for timely antimicrobial therapy leads few bovine practitioners to submit samples for bacterial culture and antimicrobial susceptibility testing in a diagnostic laboratory.

Ceftiofur is a third-generation cephalosporin with no milk withdrawal time, available for cattle administration. Several studies, which were conducted mainly in Canada, have demonstrated the efficacy of ceftiofur, both sodium and hydrochloride, and ceftiofur crystalline-free acid injectable suspension for the systemic treatment of IP in cattle, with a 99% success rate at 14 days in one study on feedlot cattle [13,14,15].

Marbofloxacin is an advanced, third-generation, veterinary fluoroquinolone approved for use in veterinary medicine in Europe and in the USA to treat respiratory, urinary and dermatological diseases that affect companion animals [16,17,18]. In Europe, it has been licenced since 1997 for use in food-producing animals (cattle and pigs) for respiratory, soft tissue and infective gastroenteric diseases; in 2000, it was registered in the UK for *E. coli* acute mastitis in dairy cattle [19,20]. 

Marbofloxacin exhibits a broad spectrum of activity, and its bactericidal activity is concentration-dependent against Gram-negative bacteria and time-dependent against Gram-positive bacteria [21,22]. Marbofloxacin’s broad spectrum of activity includes bacteria that are regularly cultured from naturally occurring septic joints in calves [23]. Recent research has shown therapeutic activity in dairy cows affected by IP [24].

Antimicrobial intravenous regional limb perfusion (IVRLP) is a well-established technique for treating or preventing orthopaedic infections of equine distal limbs [25,26]. One clinical study investigated its use for the treatment of digital septic lesions that are also found in cattle [27]. The pharmacokinetics of cefazolin [28], ceftiofur [29], florfenicol [30], tetracycline hydrochloride [31], ampicillin-sulbactam [32] and marbofloxacin [33] after IVRLP were previously defined in cattle. The IVRLP procedure can be easily performed in a field setting. This locoregional drug delivery approach requires a controlled application of an efficient tourniquet, aimed at maintaining an adequate vascular isolation in a selected portion of the distal limb, followed by the administration of the diluted drug via puncture of an accessible superficial vein. Theoretically, any antimicrobial drug that can be safely administered by the intravenous route can be selected and given by IVRLP. Currently, there are no antimicrobial drugs labelled for IVRLP in large animals. The extra-label use of fluoroquinolones and cephalosporins in livestock is banned in the USA, akin to the prohibition of the use of tetracycline hydrochloride. Although not illegal, florfenicol is not approved for use in lactating dairy cows. The extra-label use of ceftiofur and marbofloxacin is not banned in Europe, and withdrawal times are determined for meat and milk production. Nonetheless, their use as a first-choice treatment is restricted, and they should be used only when supported by a bacterial culture and by antimicrobial susceptibility test results.

The IVRLP method of antimicrobial delivery offers many advantages over systemic administration. Local administration provides particularly high antimicrobial concentrations at the site of infection and minimizes systemic diffusion and potential side effects [25,26]. A reduction of the total dose compared with animal bodyweight and minimal systemic plasma concentrations of antimicrobial agents are suitable to decrease residual values in milk of lactating dairy cattle [31].

The aim of this pilot study was to compare the clinical efficacy of a single IVRLP with marbofloxacin versus ceftiofur sodium for treating naturally occurring acute IP in dairy cows.

## 2. Materials and Methods

The clinical study was approved by the local ethical committee (Prot. 41/2012/CEISA) and was conducted in compliance with the Italian Animal Welfare guidelines.

The study was conducted in dairy farms (2 herds and 187 cows) in the province of Teramo (Italy) during the years 2014–2016 with a randomized parallel-group study design. 

The total number of animals that were enrolled in the study was 40 dairy cows (Italian Friesian) in the early lactating stage (approximately the first 15 weeks of lactation). Criteria of inclusion were a diagnosis of acute IP on the basis of history and clinical signs and a 3–4/5 locomotion score. For this study acute IP was defined as erythema and symmetric swelling of the interdigital space and the coronary band resulting in spreading of the claws, with or without evident necrotic lesion of the interdigital skin and causing acute onset of variable lameness of <36 h duration. Lameness was graded according to the scoring system devised by Sprecher et al. [34]. Exclusion criteria were concurrent disorders, medical treatment within 30 days prior to the study and clinical signs of IP or other causes of lameness in more than one limb.

Immediately after the diagnosis, the cows were randomly assigned to one of two treatment groups, marbofloxacin (M) vs. ceftiofur (C). Animals were restrained in a chute configured for claw trimming and were subjected to a standing, non-sedated single IVRLP with the selected antimicrobial drug. During the positioning of the tourniquet and drug infusion, the affected limb was on the ground in a weight-bearing position and was restrained with a rope looped around the pastern, while the opposite limb was lifted off the ground and restrained with the belt at a level immediately proximal to the hock (Dutch chute with a manual crank). Then, all four limbs were free on the ground in a weight-bearing position. To accomplish the procedure, a manual pneumatic tourniquet (7 × 35 cm cuff at 300–400 mmHg; VBM^®^ Medizintechnik GmbH, Sulz am Neckar, Germany) was placed over the proximal portion of the principal metacarpus/metatarsus, and a 19-gauge butterfly needle was introduced into the dorsal common digital III vein [24]. 

During IVRLP, group M received 0.67 mg/kg of marbofloxacin (1/3 of the daily systemic dose, Marbocyl 10%^®^ Vétoquinol, Czech Republic), while group C received 500 mg/animal of ceftiofur sodium, (Excenel^®^ RTU EZ, Zoetis, USA). For IVRLP, the antimicrobial was diluted to 40 mL with sterile water for injections and then manually infused by a slow bolus injection (over 60–90 s). After the administration of all the perfusate, as the butterfly needle was removed, a firm pressure was applied over the venipuncture site using 4 × 4 cm gauze. This compression pad was secured in place with a tape to minimize subcutaneous leakage and prevent hematoma formation while the tourniquet was left in place. The pneumatic tourniquet was released 30 min after completing the injection, and then removed.

No local (e.g., topical preparations, lesion management and bandaging) or systemic (NSAIDs) adjunctive treatment was administered. During the recovery period, all cows were housed in a clean and dry area. At every clinical re-examination, the affected limb was lifted off the ground in the trimming chute and the lesion was cleaned with soapy water and then thoroughly dried.

Clinical follow-up data (severity of lameness, digital swelling and local lesion) was obtained at 5, 10 and 15 days post-IVRLP. A bovine practitioner who was not aware of the treatment groups monitored the clinical response to antimicrobial IVRLP. Digital swelling was subjectively referred as absent, mild, moderate or marked. Interdigital skin lesions were visually recorded as absent (blind IP), healed/healing or active/non-healed open lesion. The total daily milk yield of each cow the day before the clinical appearance, the day of the diagnosis and at the clinical follow-up time points post-IVRLP was registered.

The clinical cure after treatment was considered when all the following conditions were accomplished: the disappearance of digital swelling, locomotion score reduction of at least 2/5 points, healed or healing local lesion and no relapse at the final observation (15 days post-IVRLP). 

Descriptive statistics were analysed by using Excel^®^ for Mac, version 16.72. The lameness, digital swelling and local lesion severity were compared among groups at each time-point by using the Kruskal–Wallis test followed by Dunn’s post hoc analysis, while Fisher’s exact test was performed to compare the treatment success rates among groups at 15 days. The total daily milk yield of each cow at the day before the clinical appearance, the day of the diagnosis and at the clinical follow-up time points post-IVRLP was analysed using a paired t Student test, while milk yield between the 2 treatment groups, at the same time points, was compared by a one-way analysis of variance. For all analyses, Medcalc^®^ software, version 12.5.0, was used, and a *p* value < 0.05 was considered significant.

## 3. Results

The 40 dairy cows that were included in the study were aged between 3–8 years (mean age 4.48 years ± 1.41, and median age 4 years) and bodyweight ranged between 504–578 kg (mean bodyweight 539.05 kg). 

Among the 20 animals in group M, 18 were affected on a pelvic (7 right and 11 left) and 2 on a thoracic (2 right) limb; among the 20 animals in group C, 18 were affected on a pelvic (11 right and 7 left) and 2 on a thoracic (1 right and 1 left) limb.

At the initial evaluation, all of the animals exhibited interdigital swelling and redness of the skin in the affected foot, thereby resulting in spreading of the claws, with (group C: 15/20, 75%; group M: 14/20, 70%) or without (group C: 5/20, 25%; group M: 6/20, 30%) the presence of an observable fissure or necrotic lesion in between; in both groups, 14/20 (70%) animals had a baseline locomotion score of 4/5, while 6/20 (30%) cows had a score of 3/5. Digital swelling was mild in 2/20 (10%), moderate in 10/20 (50%) and marked in 8/20 (40%) cows in group C, while being moderate in 8/20 (40%) and marked in 12/20 (60%) cases in group M. Lameness, digital swelling and local lesion severity were not significantly different between groups at any time-point during the follow up (Figure 1, Figure 2 and Figure 3). 

In both groups, by the fifteenth day after a single antimicrobial IVRLP, 17/20 (85%) cows showed a positive clinical outcome, with no significant difference (*p* > 0.05) among the groups.

All cows underwent a consistent reduction of milk yield compared to the production level on the last day before the clinical diagnosis (Figure 4; *p* < 0.0001). 

The average daily milk production resumed significantly after treatment in both groups (5th day post-IVRLP; *p* < 0.01) and continued to increase significantly at the following 2 time points (10th and 15th day post-IVRLP; *p* < 0.01), as is normal in this early lactation stage (Figure 4). Conversely, no significant difference between the 2 treatments groups was recorded for the milk yield (*p* > 0.05).

In group M, at 15 days, severe swelling and 4/5 lameness persisted in three cows (15%), as did the decrease of milk production. The same conditions occurred in three cows in group C. Of these six animals, at the initial evaluation (day 0), 6/6 (100%) had a baseline locomotion score of 4/5, 1/6 (17%) showed a moderate digital swelling and 5/6 (83%) a marked one, and 6/6 (100%) had active interdigital skin lesion. These animals were culled for economic reasons.

## 4. Discussion

In the peer-review literature, only a few studies evaluate or compare critically the clinical efficacy of systemic antimicrobial treatment of bovine IP in naturally occurring condition [13,14,15]. However, the comparison of these studies to determine the most effective drug and the clinical outcome is challenging, mainly because of the differences in the IP case definition, in the time of detection of IP cases and in the definition of treatment success.

Morck et al. compared the clinical efficacy of ceftiofur sodium at a dosage of 1.0 mg/kg IM every 24 h to oxytetracycline at dosage of 6.6 mg/kg IM every 24 h [13]. All cows assigned to the treatments group were treated for 3 days and the treatment was considered successful if animals were no longer lame on day 4 and were not re-treated for IP within 10 days after the initial treatment. These ceftiofur sodium and oxytetracycline regimens were effective for treatment of acute IP in feedlot cattle, at 73% and 68% success rates, respectively.

One clinical trial evaluated the clinical efficacy of ceftiofur sodium at a dosage of 1.1 mg/kg administered IM once daily for 3 consecutive days [14]. Clinical recovery was defined as an at least two-point reduction in lameness scores, moderate to no swelling, healed or healing lesions and no observable relapse at a final examination between day 12 and 15. The clinical recovery rate for ceftiofur sodium was 69.6% for beef cattle and 54.6% for dairy cattle.

Another study was conducted in feedlot cattle, and compared the clinical efficacy of a single injection of ceftiofur crystalline free acid sterile injectable suspension, at a dosage of 6.6 mg/kg SC in the ear, with the IM injection of ceftiofur sodium for three days, every 24 h, at a dosage of 0.02 mg/kg in the neck [15]. A success was a cow not reproposed for clinical observable IP within 14 days after treatment. Fourteen days after treatment, the recovery success rate was 99.5% in the ceftiofur crystalline free acid sterile injectable suspension group, and 99% in the ceftiofur sodium sterile powder for injection.

Torehanov et al. compared effectiveness of interventions for treating bovine IP with a meta-analysis, and found no significant differences between the risk ratios for the antimicrobials versus a placebo [35]. However, ceftiofur sodium administered intramuscularly at a dose of 1.0 mg/kg body weight every 24 h for 3 days showed a better clinical response than 6.6 mg of oxytetracycline, 2.5 mg of tulathromycin, the placebo and 0.1 mg of ceftiofur sodium [35].

To the best of our knowledge a randomized, prospective, clinical trial of IP treated with IVRLP has not been published to date.

In this field study, a single IVRLP procedure, with marbofloxacin or ceftiofur, was demonstrated to be equally safe and clinically effective for treating acute IP; indeed, the treatment success rates were high (85%) and were not significantly different between groups. Although NSAIDs could represent an adjunct therapy in the early stage of IP [2,5], in our study they were not used in order to not modify gait abnormalities and weight-shifting between the limbs in dairy cows with naturally occurring acute IP. Therefore, the use of this drug class would have altered the clinical response to single antimicrobial IVRLP and the successive evaluations during the recovery period.

Early and aggressive antimicrobial systemic treatment is recommended for acute IP, and several antimicrobial drugs have been suggested to successfully treat this condition [2,5,11,12]. Currently, in the countries where it is allowed, systemic ceftiofur administration is recognized to be the standard treatment for IP in lactating dairy cows because of the non-existent or minimal milk discard time and high success rates [36].

Unlike systemic administration, IVRLP delivers high antimicrobial concentrations to the digital region with minimal systemic diffusion. Therefore, the presence of potential violative milk residues is substantially reduced, if not absent [37]; nevertheless, in this regard, further clinical field studies with reliable and effective detection methods [38] are recommended. Furthermore, antimicrobial concentrations achieved by systemic administration are often nontherapeutic in a highly septic environment, as in the case of IP [28,29,30,31,32,33].

Moreover, higher local concentrations of the antimicrobial drug were reached with IVRLP, which is associated with a decrease in the total dose of the antimicrobial drug that is administered for treatment, and the substantial reduction of violative residues in food production (milk and meat) could reduce the dreaded potential transfer of antibiotic-resistance from food producing animals to humans [28,29,30,31,32,33,37,39].

A high success rate and a low antimicrobial concentration (1/3 or 1/2 of systemic dose, once), as well as the absence of violative residues in food production (milk), guarantees economic savings for the farmer.

A recently published review article summarizes the most important information regarding digital anatomy, infectious causes of lameness, and IVRLP in cattle [40]. There are many variations in the IVRLP procedure, including the selected antimicrobial agent and dose (concentration- vs. time-dependent), the volume and concentration of the perfusate administered (higher vs. lower volume), the ideal vessel to perfuse (dorsal common digital vein III vs. abaxial palmar or plantar digital vein III or IV for IVRLP of the digit), the type of needle used (butterfly needle vs. intravenous catheter), the type and duration of application of the tourniquet (pneumatic vs. wide rubber and shorter vs. longer time), the need for limb exsanguination and the method of restraint to reduce the limb movement (standing vs. recumbency). 

The optimal method of performing IVRLP has not been outlined; there are some contradictory evidences in the published research literature, and usually the decision is based on clinician preference. At the authors’ institution, the methodology described in this study is the preferred technique to perform antimicrobial IVRLP in a field setting. The use of the manual pneumatic tourniquet to isolate a portion of peripheral circulation, of the butterfly needle to administer the perfusate and of the trim-chute for standing restraint and immobilization of the cattle’s foot, set up this technique as relatively simple, safe and repeatable in farm animal practice.

The pharmacokinetics of some antimicrobial agents has been defined when used for IVRLP in cattle [28,29,30,31,32,33]. Antimicrobial selection for IVRLP ideally should be based on culture and sensitivity test results. Initial antimicrobial choice can be made empirically according to the clinical case characteristics and the efficacy against common bovine pedal pathogens. However, establishing an antimicrobial treatment should be warranted by the clinical evidence of digital infection, and the practitioner must choose an antimicrobial agent that can be legally administered in livestock.

An IVRLP of marbofloxacin/ceftiofur could be potentially used in Europe to treat other deep digital septic disorders in dairy cattle, but specific studies are needed to confirm the efficacy for these conditions. 

Despite the evidence of the efficacy of a single IVRLP with marbofloxacin or ceftiofur for treating IP, some limitations affect the relevance of this study, such as the relatively small sample size and the absence of a negative control group for ethical reasons. Another possible limitation of our study was the lack of a confirmatory bacteriologic test from biopsy samples of interdigital skin or lesion surface/exudate swabs.

To reduce the use of antimicrobial drug, one case-series report in dairy cows evaluated the effect of a salicylic acid claw bandage in the treatment of early-detected, non-complicated IP [41]. The advantages of this topical treatment could include reduced risk of antimicrobial resistance, no injections, cheaper treatment costs and no withdrawal time for milk. However, these benefits should be tested and confirmed in a randomized, positive-controlled, blinded study [41].

## 5. Conclusions

No published study has evaluated the effect of antimicrobial IVRLP in treating naturally occurring bovine IP. The preliminary results of the present study support the hypothesis that, in dairy cattle with an acute IP lameness, a single antimicrobial IVRLP procedure has a high success rate and restores the milk yield, regardless of the antimicrobial selected (ceftiofur vs. marbofloxacin). However, further randomized, prospective, clinical trials are needed to compare the repeatable outcomes of clinical cases of bovine IP treated with antimicrobials administered through the IVRLP technique.

Antimicrobial treatments are frequently prescribed and administered for treating lameness due to septic digital disorders in dairy cattle. In the management of IP, a responsible use of antimicrobial drugs and procedures for local antimicrobial therapy, such as IVRLP, is strongly recommended to preserve antimicrobial activity.

## Figures and Tables

**Figure 1 animals-13-01598-f001:**
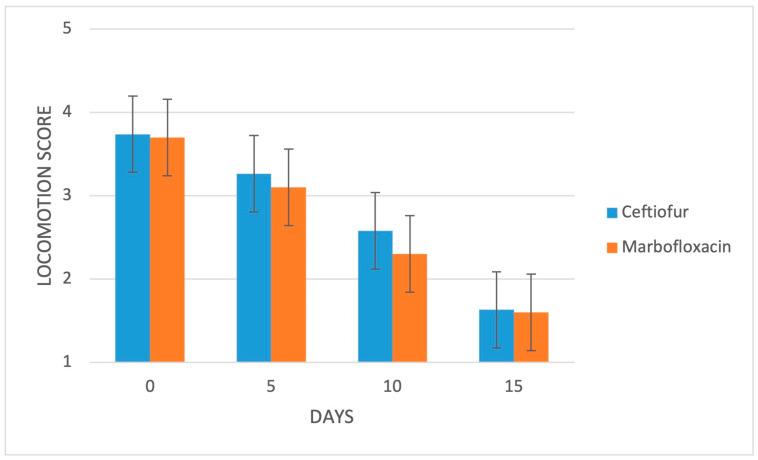
Mean locomotion scores of the 2 study groups (*n* = 20/group). Bars represent the standard error of the mean. Scores range from 1 to 5: 1 = normal, 2 = mildly lame, 3 = moderately lame, 4 = lame and 5 = severely lame [34].

**Figure 2 animals-13-01598-f002:**
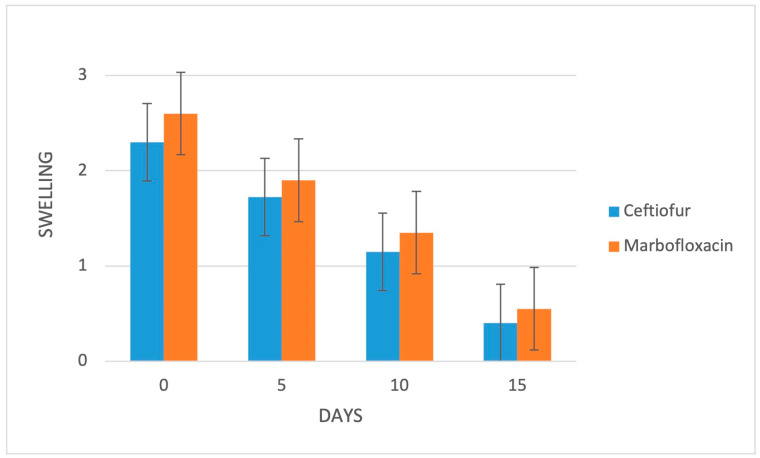
Mean swelling scores of the 2 study groups (*n* = 20/group). Bars represent the standard error of the mean. Digital swelling ranges from 0 to 3: 0 = no swelling, 1 = mild swelling, 2 = moderate swelling and 3 = marked swelling.

**Figure 3 animals-13-01598-f003:**
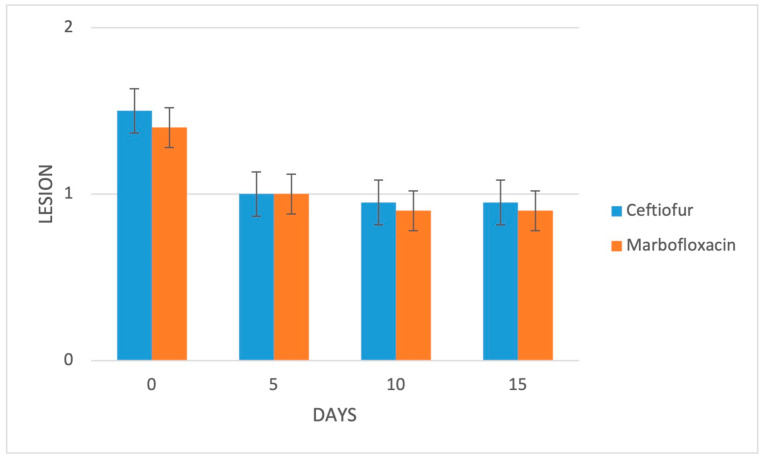
Mean local lesion severity of the 2 study groups (*n* = 20/group). Bars represent the standard error of the mean. Severity ranges from 0 to 3: 0 = no lesion, 1 = healed/healing open lesion, 2 = active/non-healing open lesion.

**Figure 4 animals-13-01598-f004:**
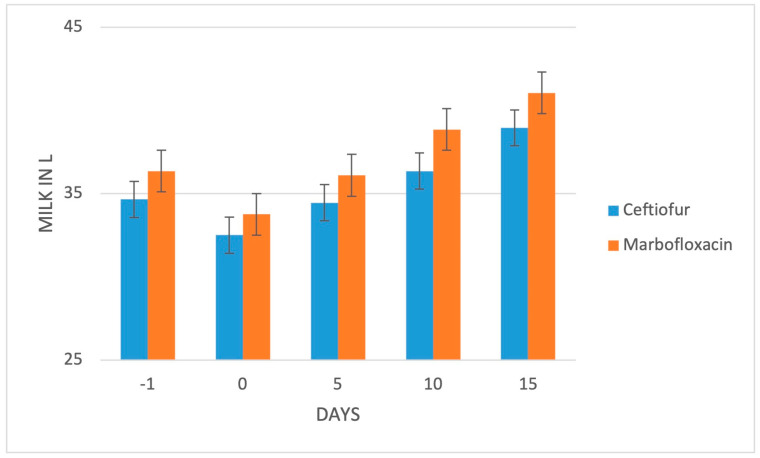
Average trend of milk yield in both treatment groups the day before the clinical appearance, the day of diagnosis, and 5, 10 and 15 days post-IVRLP. Bars represent the standard error of the mean.

## Data Availability

The data presented in this study are available on request from the corresponding author. The data are not publicly available due to privacy reasons.

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
