# Peer review of "Clinical Efficacy of a Single Intravenous Regional Limb Perfusion with Marbofloxacin versus Ceftiofur Sodium to Treat Acute Interdigital Phlegmon in Dairy Cows"

_animals, 2023, doi:10.3390/ani13101598_

Round 1

Reviewer 1 Report

Clinical efficacy of a single intravenous regional limb perfusion with marbofloxacin versus ceftiofur sodium to treat acute interdigital phlegmon in dairy cows

REMARKS AND RECOMMENDATIONS

In this interesting field study where the authors compared the clinical efficacy of a single antimicrobial intravenous regional limb perfusion (IVRLP) with marbofloxacin versus ceftiofur sodium for treatment of naturally occurring interdigital phlegmon (IP) in dairy cows. The results support the hypothesis that a single antimicrobial IVRLP procedure, irrespective of the antimicrobial selected (ceftiofur vs marbofloxacin), has a significantly high success rate and restores milk yield in cases of dairy cattle with an acute IP lameness.

In the material & method section, there is a lack of detailed information on how the intravenous application of the antibiotic on the affected limb was carried out (limb lifted and restrained in the hoof trimming chute?), information on how the clinical parameters were checked on the respective control days (0, 5, 10 and 15), and whether topical treatment and systemic treatment with NSAIDs were also carried out or not.

In addition, paragraphs with statements in the manuscript are repeatedly not supported with appropriate citations.

In the following detailed list, you can find all remarks, comments and questions that should be considered and incorporated in the revised version for improving the manuscript:

ABSTRACT

The abstract should be rephrased after revision of the complete manuscript.

INTRODUCTION

Line 41-44: Interdigital phlegmon (IP) has been known for centuries and is reported from several countries around the world. This disease has also been referred to as foot rot, foul in the foot, interdigital necrobacillosis, lure, panaritium and interdigital or infectious pododermatitis.                The authors should add some adequate references for these statements here.

Line 47-49: ….is probably less than 5%, but in epidemic outbreaks, the incidence of the disease can be as high as 20% of the milking cows in a herd [2]. The Italian prevalence is 0,73% (46 herds and 3559 cows) [3], with an incidence of 20,6% being reported in a herd of 321 dairy cows [4].        Here you listed three older references, the reference Greenough 2007 is not appropriate here, because it is not an original study, but a text-book information. The authors should add some newer data on the overall incidence of IP such as for example:

Kontturi M, Kujala M, Junni R, Malinen E, Seuna E, Pelkonen S, Soveri T, Simojoki H: (2017): Survey of interdigital phlegmon outbreaks and their risk factors in free stall dairy herds in Finland. Acta Vet Scand (2017) 59:46; DOI 10.1186/s13028-017-0313-0.

Osová A, Hund A, Mudron P (2017): Interdigital phlegmon (foot rot) in dairy cattle - an update. Wiener Tierärztl. Mschr.104, 209-220.

Tulemissova ZK, Ibazhanova AS, Myktybayeva RZ, Khussainov DM, Mussoyev AM, Kenzhebekova ZZ, Torehanoy MA (2021). Bovine interdigital necrobacillosis epizootic data from livestock farms in Almaty region of Kazakhstan between 2017 and 2019. Adv. Anim. Vet. Sci. 9(5): 761-765; http://dx.doi.org/10.17582/journal.aavs/2021/9.5.761.765.

Line 54: …. At the end of this paragraph you should add also a newer reference, e.g.:

Kontturi M, Junni R, Simojoki H, Malinen E, Seuna E, Klitgaard K, Kujala-Wirth M, Soveri T, Pelkonen S. (2019): Bacterial species associated with interdigital phlegmon outbreaks in Finnish dairy herds. BMC Veterinary Research (2019) 15:44; https://doi.org/10.1186/s12917-019-1788-x.

Line 63-64: … evident interdigital skin lesion and this condition probably represents IP in an early stage of the disease. Here at the end of this paragraph you should add some adequate references for these statements, e.g.: Van Metre 2017, Osová A, Hund A, Mudron P (2017): Interdigital phlegmon (foot rot) in dairy cattle - an update. Wiener Tieraerztliche Monatsschrift 104, 209-220.

Line 67: … and often, the cow has to be culled [6]. This one reference - a doctoral thesis - is not enough. Please add some more adequate references here such as e.g.: Van Metre 2027, Osova et al. 2017. In addition, here you should also add some information, that for more advanced (severe) cases of IP there is the possibility for a surgical debridement of all infected soft tissue in the interdigital space and digital amputation of the affected digit (in cases with subsequent purulent infection of the distal and/or proximal interphalangeal joints) before culling the cow (see Osova et al. 2017).

Line 72: … form of IP in which development of deep sepsis can occur within two days of the onset of the clinical sign. Also, here some adequate references for these statements are missing: see Osova et al 2017.

Line 74 …: Treatment of IP remains one of the main motives for therapeutic use of antimicrobial in cattle in Europe and USA. The standard treatment protocol for acute IP always includes the administration of antibiotics and NSAIDs (see Van Metre 2017, Osova et al. 2017). Please add this information here.

Line 110 …: Several clinical studies investigated its use for the treatment of digital septic lesions that are also found in cattle [23]. You are speaking from "several clinical studies", but you cite only one single study; please cite also other studies on this topic; e.g. (23-29).

Line 124-125: Nonetheless, their use as a first-choice treatment is restricted, and they should be used only when supported by a bacterial culture and by antimicrobial susceptibility test results. Yes, this is correct, but under practice conditions the vet has not the time for waiting for the results of the bacterial culture from IP (about 2 weeks for these anaerobic bacteria), PCR (some days) and for the antimicrobial test results. You should discuss this dilemma when you recommend the application of ceftiofur and marbofloxacin in dairy cattle. Did you follow this procedure in your study ? Please specify.

MATERIALS & METHODS

Should the authors add a sentence, that the present study was discussed and approved by the institutional ethics and animal welfare committee of the …. University in accordance with GSP guidelines and national legislation ???

Line 141: Criteria of inclusion were a diagnosis of acute IP on the basis of history,…: In the title you mentioned the treatment of “acute” stages of IP; therefore some information is expected how long the typical clinical symptoms of IP persisted until therapy was given in your study. Further, the authors should define "acute IP": persistence of typical clinical symptoms for 24, 48 hours …? … for how long?

Line 148: … claw trimming and were subjected to a standing, …: How the affected limb was restrained for the intravenous application in the hoof trimming chute: raised and restrained at the level proximal of the tarsal joint with a belt? Where did you apply the tourniquet? Please describe in detail.

Line 161: Here at the end of this paragraph you should add some additional information: What was done topically in the affected interdigital region? Please specify: did you clean the interdigital area and apply topical antibiotics and a bandage: yes or no? Furthermore, did you also administer NSAIDs in these patients? The administration of antibiotics and NSAIDs is the standard treatment protocol for early stages of IP (see Van Metre 2017, Osova et al. 2017). If not, please discuss why the administration of NSAIDs was withheld.

Line 163: …. obtained at 5, 10 and 15 days post-IVRLP. Did you apply some additional treatments even at days 5, 10 and 15? Please specify!

Line 165-166: Interdigital skin lesions were visually recorded as absent (blind IP), healed/healing and active/non-healed open lesion. How were these interdigital lesions checked? On the raised hind limb in the trimming chute? Please specify!

RESULTS

Line 184-185: … in the study were aged between 3–8 years 184 (mean age 4.48 years ±1.41, and median age 4 years). You should also add some information on the body weight of these cows (mean and range), because the dosage of the applied antibiotics was based on the body weight.

Figure 1: Mean lameness scores of the 2 study groups …: please replace lameness scores by locomotion scores, because score 1 describes a non-lame cow.

Figure 1 – 4: Since the authors only controlled for lameness, swelling … etc. on day 0, 5, 10, and 15, the connecting lines between these time points should be removed in figures 1 - 4. In addition, it is unclear in all the tables to which group the bars (having always the same colour) belong, which represent the standard error of the mean. Please change and clarify.

Line 236: These animals were therefore culled. What was the reason for culling these animals with "complications" or having non-positive treatment response, and why was no digital amputation performed in these animals to save them? In addition, you should add some information on the clinical signs of these 6 animals on day 0, possibly these 6 animals already suffered from an advanced (severe) stage of IP on day 0, so that IVLRP was not anymore the adequate treatment method for these cases at all.

DISCUSSION

Line 240: … systemic antimicrobial treatment of bovine IP in naturally occurring condition. Again, here are missing some adequate references.

Line 264: …. with IVRLP has not been published to date. Here in this part of the discussion you should add and discuss some results from the meta-analysis study of Torehanov MA, Tulemissova ZK, Ibazhanova AS, Rafikova ER, Muzapbarov B, Korabaev EM, Siyabekov ST (2021): Comparative effectiveness of interventions for treating interdigital necrobacillosis in cattle: A network metaanalysis. Vet Med-Czech 66, 461–469.

Line 271: … this condition [5, 7, 8]. Add here Osova et al. 2017.

Line 272-273: Currently, in the countries where it is allowed, systemic ceftiofur administration is recognized to be the standard treatment for IP in lactating dairy cows because of the non-existent or minimal milk discard time and high success rates [31]. Again, here you should add some powerful reasons why a reserve antimicrobial should be applied, when penicillin, ampicillin … etc. also promise good therapeutic results. As I said, in the daily practice it is very difficult to fulfill the guidelines of the restricted use of antibiotics, particularly in acute IP, because the vet cannot wait for some days for the results of the bacterial culture (about 2 weeks for these anaerobic bacteria) and/or the PCR result and the results of an antimicrobial test. So therefore, why the vet should not use penicillin or ampicillin for treatment of IP? Please discuss these difficult situations under practical conditions in the field.

Line 308: Antimicrobial selection for IVRLP ideally should be based on culture and sensitivity test results. Yes, this is correct, but for the practical use for treatment of acute IP the vet has not the time for waiting for the results of the bacterial culture (about 2 weeks for these anaerobic bacteria), PCR (some days) and the antimicrobial test results. You should discuss this practical problem when you recommend the application of ceftiofur and marbofloxacin in dairy cattle. The authors should present a feasible way how this situation can be decided in practice without violating the current guidelines of the use of reserve antibiotics.

Line 316-318: Possibly in the most severe, chronic IP cases, and in “non-healing” hoof lesions in dairy cows, IVRLP with antimicrobials can be repeated and could be associated with curettage/debridement of affected tissues. Sorry, this is certainly NOT a good idea: in these cases, the vet should perform a careful surgical debridement of all the infected interdigital soft tissue or in more advanced cases with a secondary purulent infection of distal interphalangeal joint, the vet should amputate the digit (see Osova et al. 2017). Repetition of IVRLP will not solve the problem at all, and furthermore a repeated IVRLP could result in other more severe complications such as a 'generalized distal limb vessel thrombosis' (see the literature).

Even for the so-called “non-healing” hoof lesions, the better and actual term is digital dermatitis-associated hoof horn lesions, repeated IVRLP will not be the adequate and evidence-based treatment option. Successful treatment methods for these longstanding, “non-healing” hoof lesions (DD-associated hoof horn lesions) were already reported, and these methods include complete removal of all the loose horn, and all the infected corium tissue by surgical debridement under local anesthesia followed by topical treatment with tetracycline spray, blocking the partner claw and applying a bandage (see the recent literature).

I suggest deleting this part of your discussion on “non-healing” hoof lesions, because this a complete other topic.

Line 323: In your discussion you should also mention and discuss the use of a non-antibiotic treatment for early cases of IP: see Persson Y, Jansson Mörk M, Pringle M, Bergsten C. (2019): A case-series report on the use of a salicylic acid bandage as a non-antibiotic treatment for early detected, non-complicated interdigital phlegmon in dairy cows. Animals 2019, 9, 129; doi:10.3390/ani9040129.

Additional interesting/important references which could be included in the manuscript:

Kontturi M, Junni R, Simojoki H, Malinen E, Seuna E, Klitgaard K, Kujala-Wirth M, Soveri T, Pelkonen S. (2019): Bacterial species associated with interdigital phlegmon outbreaks in Finnish dairy herds. BMC Veterinary Research (2019) 15:44; https://doi.org/10.1186/s12917-019-1788-x.

Kontturi M, Kujala M, Junni R, Malinen E, Seuna E, Pelkonen S, Soveri T, Simojoki H: (2017): Survey of interdigital phlegmon outbreaks and their risk factors in free stall dairy herds in Finland. Acta Vet Scand (2017) 59:46; DOI 10.1186/s13028-017-0313-0.

Osová A, Hund A, Mudron P. (2017): Interdigital phlegmon (foot rot) in dairy cattle - an update. Wiener Tieraerztliche Monatsschrift 104, 209-220.

Persson Y, Jansson Mörk M, Pringle M, Bergsten C. (2019): A case-series report on the use of a salicylic acid bandage as a non-antibiotic treatment for early detected, non-complicated interdigital phlegmon in dairy cows. Animals 2019, 9, 129; doi:10.3390/ani9040129.

Torehanov MA, Tulemissova ZK, Ibazhanova AS, Rafikova ER, Muzapbarov B, Korabaev EM, Siyabekov ST (2021): Comparative effectiveness of interventions for treating interdigital necrobacillosis in cattle: A network metaanalysis. Vet Med-Czech 66, 461–469.

Tulemissova ZK, Ibazhanova AS, Myktybayeva RZ, Khussainov DM, Mussoyev AM, Kenzhebekova ZZ, Torehanoy MA (2021). Bovine interdigital necrobacillosis epizootic data from livestock farms in Almaty region of Kazakhstan between 2017 and 2019. Adv. Anim. Vet. Sci. 9(5): 761-765; http://dx.doi.org/10.17582/journal.aavs/2021/9.5.761.765.

Reviewer 2 Report

Dear authors,

Interesting paper about a well known disorder, but to be honest not all information is relevant to my opinion

in detail:

line 36:.. procedure had a high succes rate ....

line 49: prevalence is low because the disorder is wel recognized by the farmer and treated immediately. IP is about incidence

line 76-86 can be shortened to: ... treatment of IP by penicillines and other antibiotics. However, ...

line 131: decreasing or increasing??

line 243-262 should not be part of the discussion, where the discussion is meant to compare your results with literature. So you can start with line 263. 

line 318: associated??

line 328: .. ..procedure had a high succes rate and restores the milk yield.

Round 2

Reviewer 1 Report

I made some minor suggestions on the pdf file of the revised manuscript. The autrhors shloud add these changes and add some sentences in the discussion why they did not apply NSAIDS in thease cattle patients (see my remarks).

Reviewer 2 Report

Dear authors,

no further comments, I only have some doubt if this is very practical.

Author Response

The authors would like to thank the reviewer for his/her valuable comments and suggestions.